# Compact Wideband Self-Decoupled MIMO Antenna for 5G Communications

**DOI:** 10.3390/mi13122180

**Published:** 2022-12-08

**Authors:** Qian Li, Chencheng Wan, Gangxiong Wu

**Affiliations:** 1School of Communications and Information Engineering &Artificial Intelligence, Xi’an University of Posts and Telecommunications, Xi’an 710121, China; 2Research Center for Intelligent Information Technology, Nantong University, Nantong 226019, China

**Keywords:** self-decoupled antenna, multiple input multiple output (MIMO) antenna, the fifth generation (5G), characteristic mode analysis (CMA)

## Abstract

A compact wideband self-decoupled multiple-input and multiple-output (MIMO) antenna is presented in this paper. The proposed antenna contains a pair of horizontal back-to-back elliptical tapered slots and a vertical elliptical tapered slot, which are etched on the circular metal patch. Based on characteristic mode analysis (CMA) and a suitable feeding structure, two desired characteristic modes (CMs) are excited. Therefore, across the entire matched bandwidth, a high level of isolation is realized without external decoupling structures. For validation, a prototype is fabricated and measured, and the measured results demonstrate that an impedance bandwidth of 3000 MHz with isolation higher than 20dB is achieved. Due to its self-decoupled property, high isolation, wide bandwidth, and compact size, the proposed antenna has excellent potential for 5G antenna array applications.

## 1. Introduction

The fifth generation (5G) technology can provide a high throughput rate and a low latency in modern wireless communication systems. As one key technology of 5G communications, MIMO technology can significantly improve the channel capacity without increasing the spectrum resources and antenna transmission power. However, mutual coupling between MIMO antenna elements will severely degrade the system performance. In order to reduce the coupling between antenna elements, researchers have proposed a variety of technical schemes [1,2,3,4,5,6,7,8,9,10,11,12,13,14,15,16,17,18,19], which can be divided into the following two categories. The first category is of schemes that cancel out the original coupling between antenna elements by introducing a new coupling path. Typical structures, including parasitic elements and neutralization lines [1,2,3,4,5], have been widely used. The second decoupling category consists of schemes that suppress the propagation of surface waves by introducing some band-stop structures, such as artificial structures and resonators [6,7,8,9,10,11,12]. However, the two decoupling techniques mentioned often require additional decoupling structures. The extra space required for the decoupling structures will increase the system complexity, making it less appealing.

Focusing on the shortcomings of the technical schemes mentioned above, some self-decoupled techniques have been investigated in [13,14,15,16,17,18,19,20,21]. A self-decoupled antenna array was presented in [13], which utilized the cancellation of two opposite couplings to realize high isolation. Using the coupling cancellation of anti-phase currents, high isolation was achieved between two symmetrically placed differential mode (DM) and common mode (CM) antennas in [14]. In [15], a tightly arranged antenna-pair with low mutual coupling was realized using an orthogonal-mode method. In [20], it was found that by properly placing the antenna element in the weak-field area of the adjacent antenna element, low mutual coupling can be achieved. However, most of these works focused on the research of mobile phone antenna and the isolation bandwidth is narrow. Furthermore, the CMA can provide a clear physical insight for the operating principle of antenna, but it is not effectively used in the antenna decoupling methods mentioned above.

In this paper, we present a compact wideband self-decoupled MIMO antenna for 5G communications. The CMA is performed to offer a deep physical insight of the self-decoupling mechanism. Based on the orthogonal characteristics, a circular patch etched with a pair of horizontal elliptical tapered slots and a vertical tapered slot is introduced. The bent microstrip feeding line for the pair of horizontal elliptical tapered slots excites the mode, which is orthogonal to that of the vertically etched elliptical tapered slot. Thus, high isolation can be achieved in a wide frequency band without any additional decoupling structures. This article is organized as follows: Section 2 describes the design and analysis of the proposed self-decoupled antenna. In Section 3, the simulated and measured results are analyzed. Finally, Section 4 draws our conclusions.

## 2. Design and Analysis of the Self-Decoupled Antenna

### 2.1. Characteristic Mode Analysis of the Self-Decoupled Antenna

According to the theory of characteristic mode (TCM) [22], the current flowing on an arbitrary perfect electric conductor (PEC) can be decomposed into sets of CMs, and the CMs can be defined by:(1)$$X\left(J_{n}\right)=\lambda_{n}R\left(J_{n}\right)$$
where X and *R* represent the imaginary and real part of the generalized impedance matrix. λ*_n_* and J*_n_* are the eigenvalue and characteristic current of mode *n*, respectively [23]. It should be noted that λ*_n_* is an important parameter because it reveals resonance information and contribution to the radiation of corresponding CMs. Additionally, mode significance (MS), as an alternative parameter of λ*_n_*, is preferred. When a CM resonates, λ*_n_* is equal to 0 and the MS is equal to 1. MS can be expressed as follows:(2)$${\mathrm{MS}}_{n}=\left|\frac{1}{1+j\lambda_{n}}\right|$$

In order to investigate the self-decoupled mechanism of the proposed two port MIMO antenna shown in Figure 1, a CMA is performed using electromagnetic simulation software with an integral equation solver. Due to the limitation of the solver, the feed port and dielectric substrate are removed in the CMA, and the material and the thickness of the antenna structure are set to be PEC and 0 mm, respectively.

The predicted MSs of the first six CMs of the self-decoupled antenna are given in Figure 2. From the traces of MSs, it can be seen that CMs (mode 1, mode 2, mode 4, and mode 5) are much easier to excited. Moreover, the MSs of mode 4 and 5 are close to 1 in a wider frequency range than other modes, which indicates that the two modes have a wider potential bandwidth.

Figure 3 shows the current distributions of the most relevant CMs (mode 1, 2, 4, and 5) at their corresponding potential resonant frequencies. As shown, the current of mode 2 concentrates on the edges of the vertically etched elliptical tapered slot. The current of mode 4 mainly distributes on the edges of the pair of back-to-back elliptical tapered slots, the amplitude of the current is approximately equal, and the phase is opposite. The current of mode 1 and 5 distributing on the edges of the pair of horizontal back-to-back elliptical tapered slots have the same phases and approximately equal amplitudes. Still, the current direction reverses on the edges of the two back-to-back tapered slots.

The full wave simulated current distributions on the proposed self-decoupled antenna for different excitation signal is presented in Figure 4. As shown in Figure 4, the current distributions fed by port A and B, respectively, are similar to those of mode 2 and mode 4 shown in Figure 3, which also verifies that mode 2 and mode 4 were excited successively. Based on the orthogonality between the vertical tapered slot mode and the horizontal tapered slot mode, a satisfying level of isolation is obtained.

### 2.2. Proposed Antenna Configuration

Figure 1 illustrates the configuration of the proposed self-decoupled antenna. A 0.508 mm thick Rogers RO4350B with εr = 3.66 and tanσ = 0.0037 is chosen as the substrate board. An elliptical tapered slot and a pair of back-to-back elliptical tapered slots are etched orthogonally in the circular patch. It is worth mentioning that the elliptical tapered slot has the same configuration as in [24]; the proposed self-decoupled antenna, by contrast, has the properties of being light weight, low profile, and has compatibility with microwave circuits. The feeding network consists of an L-shape slot and a corresponding stepped microstrip line with distinct widths, which is used for realizing miniaturization and improving impedance matching. The dimension parameters are given in Table 1.

## 3. Results

To confirm the performance of the proposed self-decoupled antenna, a practical antenna model is simulated, fabricated and measured. The simulated and measured results of the S-parameter results are compared as illustrated in Figure 5, in which a photograph of the practical antenna model is displayed. From the results, we can conclude that the simulated and measured reflection coefficients excited separately by different ports were both lower than −10 dB across 3 to 6 GHz, and isolation levels higher than 20 dB between the two ports were also obtained. The 3-D radiation patterns of the proposed antenna at 4.5 GHz excited separately by distinct ports are displayed in Figure 6, as it can be seen that the maximum gains of the proposed antenna when excited separately by distinct ports point in different directions. Figure 7 depicts the measured realized peak gain when port A and port B are excited, respectively. As observed from Figure 7, over the working band, the measured realized peak gains fed through port A varies between 1.93 and 3.11 dBi, respectively, while that fed through port B varies from 2.82 to 4.81 dBi, respectively.

In order to evaluate the diversity performance of the proposed self-decoupled antenna, the envelope correlation coefficients (ECCs) are calculated using the method in [25], as shown in Equation (3):(3)$$\rho_{12}=\frac{{\left|\iint_{4\pi }\left[{\vec{E}}_{1}(\theta ,\varphi )\cdot {\vec{E}}_{2}(\theta ,\varphi )\right]d\Omega \right|}^{2}}{\iint_{4\pi }{\left|{\vec{E}}_{1}(\theta ,\varphi )\right|}^{2}d\Omega \iint_{4\pi }{\left|{\vec{E}}_{2}(\theta ,\varphi )\right|}^{2}d\Omega }$$

The calculated and measured ECCs of the proposed antenna using Equation (3) are shown in Figure 8. Slight discrepancies between calculated and measured results can be observed, which may be a result of fabrication and measurement tolerances. From Figure 8, we can see that the measured ECC values are below 0.03 across 3–6 GHz, which indicates a satisfactory diversity performance is obtained. 

In order to highlight the advantages of the proposed design scheme, Table 2 provides the performance comparisons between our decoupling scheme and several recently reported designs. The antennas in [13,14,20] only cover a narrow 20 dB isolation bandwidths. Although the property of wide decoupling is realized in the antenna in [19], it suffers from relatively poor port isolation. In comparison, the proposed self-decoupled design has the advantages of simple structure, compact size, and wide 20-dB isolation bandwidth, making it appealing for modern 5G applications.

## 4. Conclusions

In this article, a dual-port compact wideband MIMO antenna with a self-decoupled property is presented. CMA is used to explain the decoupling mechanism of the design scheme, and high isolation is obtained due to the orthogonality of the two excited CMs. The isolation performance is better than 20 dB across the desired band of 3–6 GHz. In addition, the ECC of the proposed MIMO antenna is below 0.03. The proposed self-decoupled antenna combines the merits of compact size, broad bandwidth, and high isolation, which make it a prospective candidate for applications of 5G communication systems.

## Figures and Tables

**Figure 1 micromachines-13-02180-f001:**
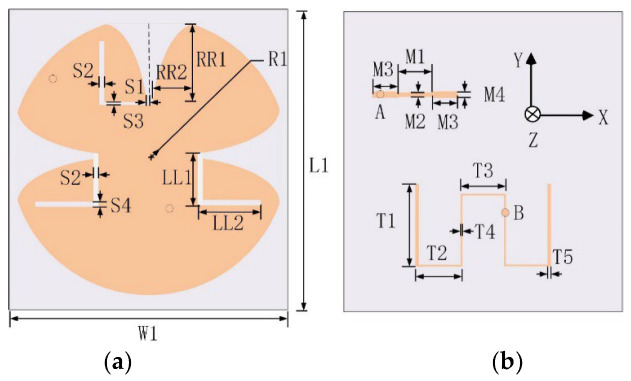
Schematic diagram of the proposed design: (**a**) front view, and (**b**) back view.

**Figure 2 micromachines-13-02180-f002:**
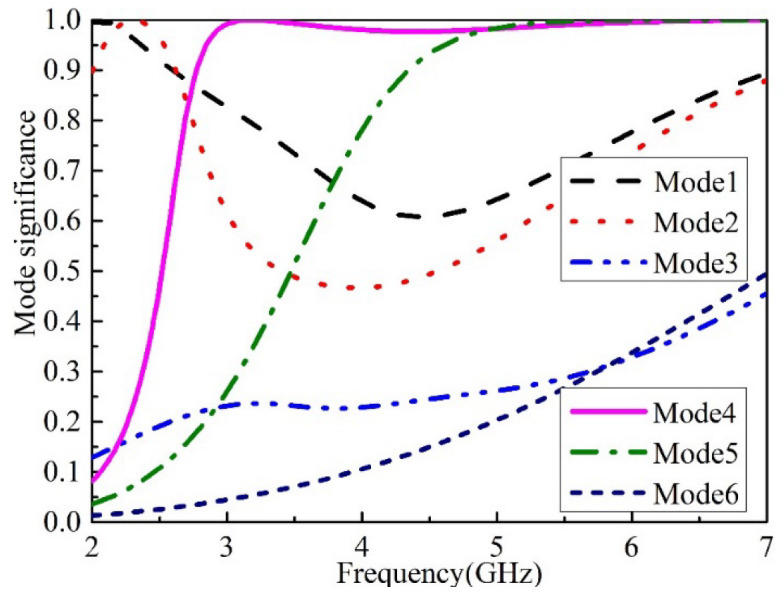
MSs of the first six CMs of the proposed antenna.

**Figure 3 micromachines-13-02180-f003:**
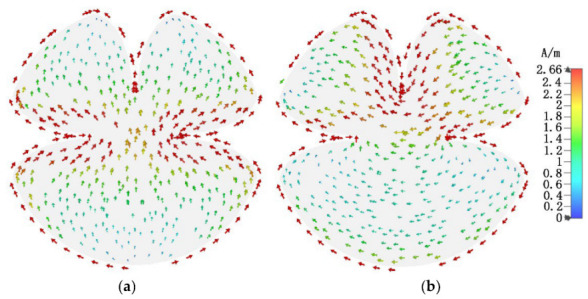
Current distributions of the most relevant CMs: (**a**) mode 1 at 2 GHz, (**b**) mode 2 at 2.3 GHz, (**c**) mode 4 at 3.17 GHz, and (**d**) mode 5 at 6 GHz.

**Figure 4 micromachines-13-02180-f004:**
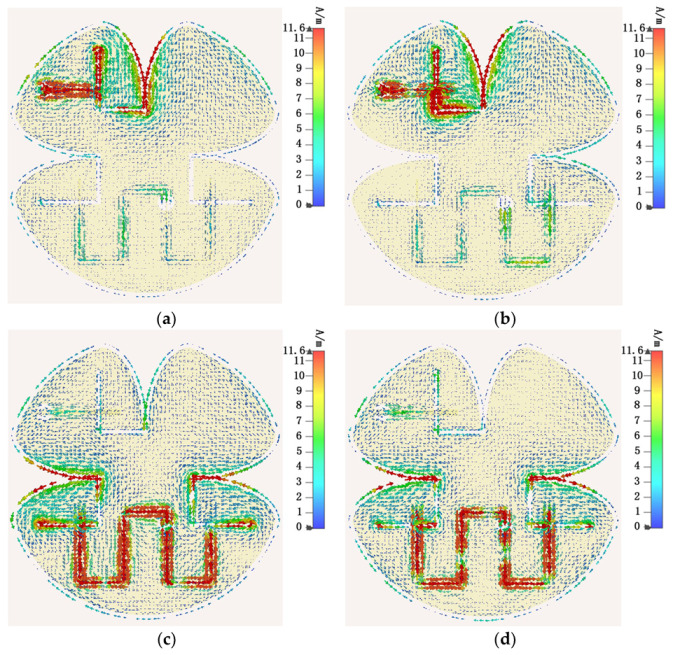
Predicted surface currents of the proposed design: (**a**) fed through port A at 4.5 GHz, (**b**) fed through port A at 5.5 GHz, (**c**) fed through port B at 4.5 GHz, and (**d**) fed through port B at 5.5 GHz.

**Figure 5 micromachines-13-02180-f005:**
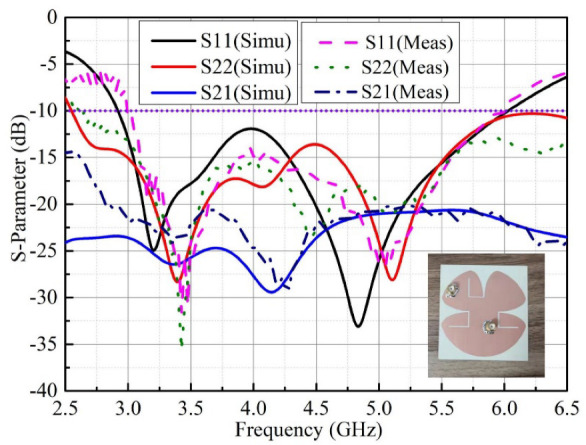
Simulated and measured results of S-parameters for the proposed self-decoupled antenna.

**Figure 6 micromachines-13-02180-f006:**
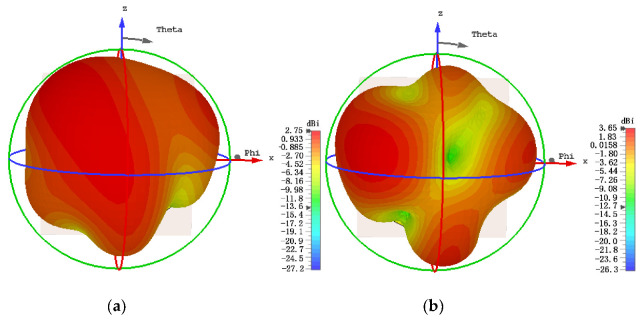
The 3-D radiation patterns of the proposed antenna at 4.5 GHz excited separately by distinct ports: (**a**) port A and (**b**) port B.

**Figure 7 micromachines-13-02180-f007:**
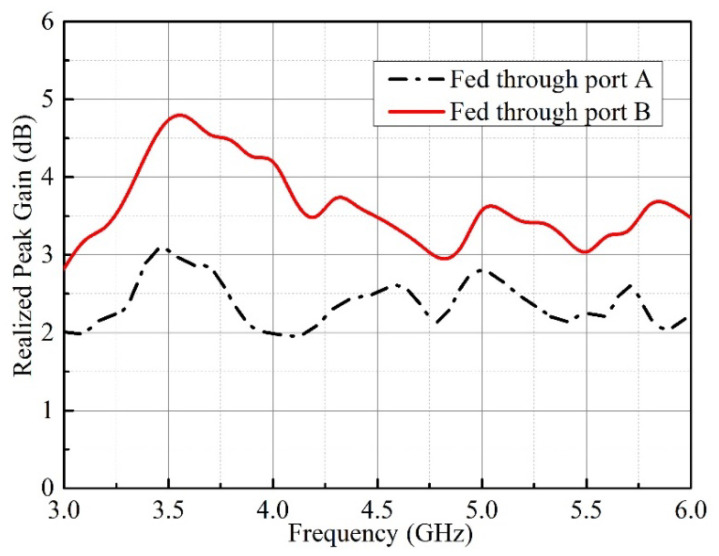
Measured realized peak gain of the proposed design.

**Figure 8 micromachines-13-02180-f008:**
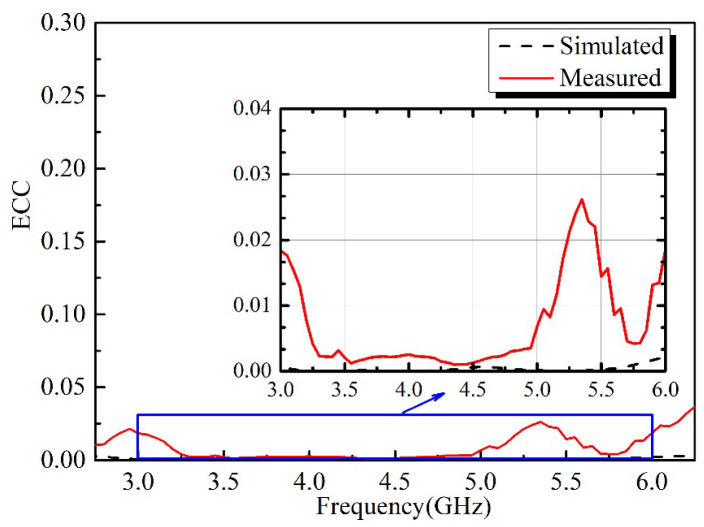
Simulated and measured ECCs of the proposed antenna.

**Table 1 micromachines-13-02180-t001:** Dimensions of the proposed antenna.

Parameter	Value (mm)	Parameter	Value (mm)	Parameter	Value (mm)	Parameter	Value (mm)
L1	70	W1	65	R1	32.5	RR1	20.3
RR2	10	S1	0.55	S2	1	S3	0.75
S4	1	LL1	11	LL2	14	M1	8
M2	1	M3	6	M4	1.5	T1	19.3
T2	10.5	T3	9.9	T4	0.3	T5	0.5

**Table 2 micromachines-13-02180-t002:** Comparisons between the proposed dual-port MIMO antenna and previous works.

Ref.	Isolation Level	20-dB Isolation BW	Structure Complexity	Volume (λ_0_^3^)	Decoupling Schemes
[13]	>20 dB	14%	Moderate	1.17 × 0.7 × 0.019	Coupling cancellation
[14]	>20 dB	5.5%	Simple	0.33 × 0.058 × 0.019	Modes cancellation
[19]	10.8 dB	0%	Moderate	0.39 × 0.097 × 0.025	Connecting line
[20]	>20 dB	<5%	Simple	1.2 × 0.7 × 0.037	Weak-field
Our work	>20 dB	67%	Simple	0.975 × 1.065 × 0.008	Different CMs

## Data Availability

The data supporting the findings of this study can be made available to genuine readers after contacting the corresponding authors.

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
