# Peer review of "Compact Wideband Self-Decoupled MIMO Antenna for 5G Communications"

_micromachines, 2022, doi:10.3390/mi13122180_

Round 1
Reviewer 1 Report
This manuscript presents a wideband MIMO antenna which achieve a good performance. The CMA is applied to analyzed the working modes and the high isolation is achieved with the suitable feeding structure. In this reviewer’s opinion, the antenna performance is acceptable. However, some minor revisions should be made as follows.
1. The slot structure is a decoupling method which might be applied in this manuscript. Thus, [Ref A-Ref B] should be added in the literature review.
[Ref A] Y. Wang, Zhengwei Du, “A Wideband Quad-Antenna System for Mobile Terminals,” AWPL, 2014.
[Ref B] ] S. Zhang, B. K. Lau, Y. Tan, Z. Ying, and S. He, “Mutual coupling reduction of two PIFAs with a T-shape slot impedance transformer for MIMO mobile terminals,” IEEE Trans. Antennas Propag., Mar. 2012.
2. Color bar should be added in Fig. 4 and Fig. 6.
3. It seems that the -10 dB impedance bandwidth is wide in Fig. 5. So, the simulated and measured frequency band should be wider than 3-6 GHz.
4. How about the antenna efficiency? It is suggested to calculate the ECC from the radiation patterns in Fig. 8.
Reviewer 2 Report
The authors should consider the following comments:
1- In Page 6, better to list the antenna dimensions in a Table.
2- It is not clear, how the antenna dimensions are obtained. The author should explain this carefully.
3- Figure 1 is cited after Figure 4. The author should manage the figures presentation in a sequence.
